# The Genetic Mechanisms Underlying the Concerted Expression of the *yellow* and *tan* Genes in Complex Patterns on the Abdomen and Wings of *Drosophila guttifera*

**DOI:** 10.3390/genes14020304

**Published:** 2023-01-24

**Authors:** Komal K. B. Raja, Evan A. Bachman, Catrina E. Fernholz, David S. Trine, Rebecca E. Hobmeier, Nathaniel J. Maki, Timothy J. Massoglia, Thomas Werner

**Affiliations:** 1Department of Pathology & Immunology, Baylor College of Medicine, Houston, TX 77030, USA; 2College of Human Medicine, Michigan State University, 965 Fee Rd. A110, East Lansing, MI 48824, USA; 3Department of Internal Medicine, University of Michigan Medical School, Ann Arbor, MI 48109, USA; 4Department of Biological Sciences, Michigan Technological University, 1400 Townsend Dr., Houghton, MI 49931, USA; 5Medizinische Fakultät, Friedrich-Alexander-Universität Erlangen-Nürnberg, Krankenhausstraße 12, 91054 Erlangen, Germany

**Keywords:** wingless, abdominal-A, hedgehog, zerknüllt, mad, cubitus interruptus, engrailed, *Drosophila*, *Drosophila melanogaster*, *Drosophila guttifera*, tan, yellow, CRM, CRE, cis-regulatory, cis-regulatory module, cis-regulatory element, evo-devo, evolution, development, pigmentation, insect, melanin, in situ hybridization, DsRed, enhancer, transgenic, reporter, expression, abdomen, wing, fruit fly, fly, spot pattern, polka-dotted pattern, co-expression, co-evolution

## Abstract

How complex morphological patterns form is an intriguing question in developmental biology. However, the mechanisms that generate complex patterns remain largely unknown. Here, we sought to identify the genetic mechanisms that regulate the *tan* (*t*) gene in a multi-spotted pigmentation pattern on the abdomen and wings of *Drosophila guttifera*. Previously, we showed that *yellow* (*y*) gene expression completely prefigures the abdominal and wing pigment patterns of this species. In the current study, we demonstrate that the *t* gene is co-expressed with the *y* gene in nearly identical patterns, both transcripts foreshadowing the adult abdominal and wing melanin spot patterns. We identified *cis*-regulatory modules (CRMs) of *t*, one of which drives reporter expression in six longitudinal rows of spots on the developing pupal abdomen, while the second CRM activates the reporter gene in a spotted wing pattern. Comparing the abdominal spot CRMs of *y* and *t*, we found a similar composition of putative transcription factor binding sites that are thought to regulate the complex expression patterns of both terminal pigmentation genes *y* and *t*. In contrast, the *y* and *t* wing spots appear to be regulated by distinct upstream factors. Our results suggest that the *D. guttifera* abdominal and wing melanin spot patterns have been established through the co-regulation of *y* and *t*, shedding light on how complex morphological traits may be regulated through the parallel coordination of downstream target genes.

## 1. Introduction

How complex body forms and patterns develop is a central question in biology [1,2]. Until the early 1970s, it was thought that very different genes underlie the morphological diversity observed in different metazoan species. However, the investigation of genomic sequences from several organisms, such as the fruit fly [3], mouse [4], and human [5], has led to the surprising discovery that many animals share most of their genes, an important and highly conserved set of which is known as the “toolkit genes”. Toolkit genes are developmental genes that regulate the building of animal bodies [6,7]. Changes in the expression of these genes underlie evolutionary novelties, e.g., beetle horns [8] and insect color patterns [9]. Besides building metazoan body morphologies, a few examples are known in which toolkit genes have been co-opted into pigmentation pathways, where they play crucial roles as regulators of melanin pigmentation. For example, *optix* [10], *spalt* (*sal*) [11], *Distal*-*less* (*Dll*) [12], *wnt*-*A* [13], and *Antennapedia* (*Antp*) [14] were shown to regulate the wing pigment patterns of some butterfly species. Similarly, previous studies have demonstrated that toolkit genes regulate pigment patterning on the wings and bodies of several *Drosophila* species. For instance, the genes *Dll* [15] and *engrailed* (*en*) [16] shape the black pigment spot on the wings of *Drosophila biarmipes*, while *wingless* (*wg*) induces black melanin spots on the wings and possibly abdomen of *D. guttifera* (*D*. *guttifera*) [1,2]. Furthermore, several toolkit genes, including *Abdominal-B* (*Abd-B)*, *doublesex* (*dsx*)*,* and *bric*-*a-brac 1* (*bab1*), were shown to regulate male-specific abdominal pigmentation in *Drosophila melanogaster* [17,18].

The *tan* (*t*) gene is an important player in insect pigmentation, and studies involving *t* offer an excellent opportunity to investigate the mechanisms that regulate color pattern formation and complexity [19]. The *t* gene product is required for the production of brown melanin [20,21] and is sometimes co-expressed with *yellow* (*y*) [22,23,24]. Several studies have demonstrated that *t* expression patterns foreshadow melanin patterns on butterfly wings [25] and caterpillars [26], suggesting that *t* expression changes can drive the evolution of animal color patterns across insects. Independently acting *cis*-regulatory modules (CRMs) of the *t* gene have been previously identified, which control its expression in different body parts and tissues [27,28]. Changes in the CRMs that regulate *t* have been implicated in the evolution of color patterns in various *Drosophila* species. For instance, it has been shown that non-coding changes in the *t* locus contribute to the abdominal pigmentation variation between the two closely related *Drosophila* species *D. americana* and *D. novamexicana* [29]. In another study, it was shown that evolution in a *t* CRM caused the divergence of abdominal pigmentation between *D*. *yakuba* and *D*. *santomea* [27]. Using Genome-Wide Association Studies (GWAS), it was demonstrated that regulatory changes near the *t* gene underlie variation in abdominal pigmentation in several *D*. *melanogaster* populations [30,31,32]. These studies suggest that *t* is an important player in the evolution of morphological diversity in *Drosophila* species. Most interestingly, *t* expression has co-evolved with *y* expression [22,23], where both genes are co-expressed in the developing adult cuticle of diverse *Drosophila* species [24]. Although these two genes have been known to be co-expressed over considerable evolutionary time [22,27], very few studies have demonstrated how two or more genes are transcriptionally coupled and regulated. For instance, it was shown that the four neurogenic ectoderm CRMs of *D. melanogaster* consist of binding sites, to which a shared set of transcription factors bind and regulate the concerted gene expression of these CRMs in the embryonic neuroectoderm [33,34]. Similarly, a recent study on simple abdominal pigmentation of some *Drosophila* species belonging to the subgenus *Sophophora* has demonstrated that the coordinated expression of the *y* and *t* genes is due to different regulatory inputs [22,23]. However, nothing is known about the genetic mechanisms that keep *y* and *t* co-expressed in complex patterns, such as in species belonging to the quinaria group [35]. 

*D. guttifera*, a member of the quinaria species group, is splendidly adorned with complex color patterns on its wings, thorax, and the abdomen [1,2,35,36,37]. The pigmentation pattern of this species provides an excellent opportunity to identify the regulatory logic that controls the expression of two genes in parallel [24]. The abdominal pigment pattern consists of distinct pattern elements: a dorsal (d), median (m), and lateral (l) pair of longitudinal spot rows, and a dorsal midline shade (dms) (Figure 1A,B). In addition to spots and a dorsal midline shade, the abdomen also displays a distinct shade that surrounds the spotted regions as well as thin black stripes along the intersegmental boundaries [24]. Similarly, the wings of *D. guttifera* display 16 melanin vein spots and four intervein shades. Previously, we showed that *y* gene expression precisely foreshadows the adult abdominal [1] and wing [2] melanin patterns. We identified one *y* CRM that drives reporter expression in six longitudinal spot rows on the abdomen. The *y* spot CRM reveals a stripe CRM at its core that drives stripes on the abdomen parallel to each segmental boundary. We also identified that the genes, *decapentaplegic* (*dpp*), *wg*, *hedgehog* (*hh*), *abdominal-A* (*abd-A*) and *zerknüllt* (*zen*) may collectively induce *y* to give rise to the abdominal melanin pattern. In the current study, we show that the *t* gene is largely co-expressed with *y* in a spotted pattern on the abdomen of *D. guttifera*. We identified independent CRMs of the *t* gene, one of which drives the reporter in six longitudinal spot rows on the abdomen. We further found that the putative transcription factor binding site content of this CRM is similar to the *y* CRM that controls *y* expression in an identical abdominal spot pattern, suggesting that both *y* and *t* are regulated by the same *trans*-factors. These results bear on our understanding about how developmental pathways generate novel morphological features in animals.

## 2. Materials and Methods

### 2.1. Fruit Fly Stocks and Genomic DNA

The fly stocks that we used are the *D*. *melanogaster* fly strain VK00006 (cytogenic location 19E7), and *D*. *guttifera* (stock number 15130-1971.10). We extracted the genomic DNA using a squish method described in [38]. The DNA was purified by using Genomic tip-20/G columns from Qiagen.

### 2.2. RNA In Situ Hybridization

We made a *t* RNA probe by amplifying partial protein-coding sequences of the *t* gene using PCR from genomic DNA. Using TA cloning, we inserted the PCR product into the pGEM-TEasy vector. We then made DIG-RNA probes by re-amplifying the PCR products from the vector, followed by in vitro transcription. The pupal wings and abdomens were dissected and subjected to hybridization with the probe (1:500) as described previously [27,39]. We stained the pupae using NBT and BCIP solutions from Promega. For each in situ hybridization experiment, we analyzed at least 30 pupal wings and 30 abdominal halves for each developmental stage (P6 to P15) [39]. The pupal wings and abdomens that developed a staining pattern were imaged using an Olympus SZX16 stereo microscope with an Olympus DP72 digital camera.

### 2.3. PCR Primers

We amplified the CRMs from genomic DNA using the following forward and reverse primers:*gut t wing* CRM: Forward- 5’-GCAGCATCGACAGTCGAGTTA-3’ and Reverse: 5’-CAACAAACGCTACGTTGACC-3’.*gut t body* CRM: Forward: 5’-GCTTACCATATCGAAGCCGAC-3’ and Reverse: 5’-GGAAGTTGAACTTCCATAACTCG-3’.*gut t wing spot* CRM: Forward: 5’-CTCGTTATAGGCGAGTGCCAAT-3’ and Reverse: 5’-AGATGTGCAAAAGTCCCACG-3’.*gut t abdominal spot* CRM: Forward: 5’-CGGTGGAGTATGGTGATTAAAG-3’ and Reverse: 5’-AACAGCCGATTCGATATAGC-3’.*gut t core stripe* CRM: Forward: 5’-GCGTGCACTTAATTGTCCAAC-3’ and Reverse: 5’-AACAGCCGATTCGATATAGC-3’.

We used KpnI and SacII restriction sites for all forward and reverse primers, respectively. 

### 2.4. Transgenic Assays

In order to identify the CRMs, we designed reporter constructs by amplifying DNA fragments of the non-coding regions of the *t* gene, using Phusion polymerase (ThermoFisher). We sub-cloned the purified PCR products into the shuttle vector *pSLfa1180fa* [40], which harbors the *DsRed* gene. These reporter cassettes were cleaved from the shuttle vector and inserted into the *piggyBac* vector (*pBac{3xP3-EGFPaf}*) [41]. The resultant *piggyBac* plasmids were injected into *D. guttifera* embryos, along with the helper plasmid “*phspBac*” to produce stable transgenic lines, as described in [42]. To generate transgenic *D*. *melanogaster* lines, we used the vector *pS3aG* for site-specific integration of transgenes [18]. The transgenic pupal wings and abdomens expressing the DsRed CRM activity were dissected out from their pupal cases and mounted in Halocarbon 700 oil for imaging. We generated multiple images of the specimens by imaging the samples at different focal depths (z-stacking), using an Olympus SZX16 stereo microscope (RFP filter set: Ex540-580/Em610). These images were processed using Helicon Focus Software to generate composite images.

## 3. Results

### 3.1. t Gene Expression Foreshadows the Adult Abdominal Melanin Pattern of D. guttifera

Previously, we showed that the *y* gene is expressed at specific sites on the developing *D*. *guttifera* pupal abdomen, precisely prefiguring the adult abdominal melanin spots and shades [1]. Using RNA in situ hybridization, we tested if the expression pattern of *t* correlates with the adult abdominal melanin pattern. Because *y* expression starts at pupal developmental stage P10, we tested *t* expression on the abdomen from the pupal stages P10 to P15. Our in situ hybridization experiments reveal that *t* expression begins at P11 and perfectly foreshadows the adult abdominal spot pattern. We found that *t* mRNA was expressed in the dorsal, median, and the lateral spot rows (Figure 1B,D and Appendix A). However, we did not observe any shade expression along the dorsal midline or around the spotted regions, as well as no strong expression at the intersegment boundaries. 

### 3.2. The CRM Controlling the Abdominal Spot Pattern of t Is Located in the Third Intron of t

Based on our in situ hybridization data [1] of putative *trans*-factor genes, we hypothesized that there may be at least three independently acting CRMs of the *t* gene that induce individual subsets of the entire abdominal spot pattern. To identify these CRMs, we transformed *D. guttifera* with constructs containing the non-coding regions of the *t* gene fused to DsRed (Figure 2A). We identified a 2-kb fragment within the third intron of the *t* gene that drove a DsRed reporter expression pattern in six longitudinal spot rows on the adult abdomen. We termed this fragment the *gut t abdominal spot* CRM (Figure 2A,C). We additionally identified several other tissue-specific CRMs of the *t* gene, such as the *gut t wing spot* CRM (more detail on this is presented later) (Figure 2A,B), the *gut t wing* CRM (Figure 2A,D), and the *gut t body* CRM (Figure 2A,E). We did not find a *t* CRM that accounts for the fainter pigment shade on the abdomen in our transgenic assay. Our findings suggest thus far that the abdominal *t* spot expression pattern is controlled by only one CRM located in the third intron of the *t* gene.

### 3.3. The Dissection of the Gut t Abdominal Spot CRM Reveals a Core Stripe CRM

We next sought to understand if the *gut t abdominal spot* CRM can be further broken down into independently acting CRMs or if the CRM is similar to the *gut y abdominal spot* CRM, which when subdivided, showed a single stripe CRM at its core that drove horizontal bands along the posterior edges of each abdominal segment [1]. Considering these two possibilities, we dissected the *gut t abdominal spot* CRM into multiple overlapping sub-fragments (Figure 3). We first subdivided the *gut t abdominal spot* CRM into two fragments: one 1258-bp left sub-fragment and a 1146-bp right sub-fragment. We observed that the left sub-fragment showed no reporter activity, whereas the right sub-fragment displayed horizontal stripe activity on the posterior part of each abdominal segment, particularly on the lateral parts of the abdomen (#1 and #2 in Figure 3). These data suggest that *t* is induced in the shape of horizontal stripes on the posterior part of each abdominal segment by the right half sub-fragment. To narrow down the *gut t abdominal spot* CRM to its core CRM sequence, we dissected the *gut t abdominal spot* CRM into four overlapping sub-fragments: a 5’ 766-bp fragment, a 719-bp fragment adjacent to the 5’ fragment, a third 964-bp fragment, and a fourth 3’ 800-bp fragment (#3, #4, #5 and #6 in Figure 3). To our surprise, none of these fragments showed DsRed reporter activity, which suggests that the core CRM sequences required to drive *t* in six abdominal spot rows are present in sub-fragment #2, which lost CRM activity upon further sub-fragmentation. Our data revealed a stripe element at the core of the *gut t abdominal spot* CRM; we thus termed fragment #2 the *gut t core stripe* CRM.

### 3.4. The Gut t Abdominal Spot CRM Is Inactive in Transgenic D. melanogaster

In order to test if the transcription factor landscape of *D. melanogaster* would activate the *gut t abdominal spot* CRM and its sub-fragments, we transformed the *D. guttifera*-derived CRM and its sub-fragments into *D. melanogaster*. Because the *trans*-landscapes of *D*. *guttifera* and *D*. *melanogaster* likely have changed over the past 60 million years of divergence [43], the reporter expression patterns observed in *D*. *melanogaster* would allow for the identification of the *trans*-factors that activate the *D*. *guttifera* CRMs. This would be possible by comparing the resulting reporter expression patterns with known toolkit gene expression patterns in the *D*. *melanogaster* abdomen. However, when examining the transgenic *D*. *melanogaster* pupal abdomens for reporter activity throughout development, we found that the *gut t abdominal spot* CRM and its sub-fragments were inactive in the *trans*-landscape of *D*. *melanogaster*. This means that the CRM and its sub-fragments are not recognized by the transcription factor landscape of *D*. *melanogaster*, suggesting a stark change in the *trans*-landscape between *D. guttifera* and *D. melanogaster* over the past ~60 million years. In the opposite direction, we were unable to transform *D*. *guttifera* with *D*. *melanogaster* sequences because the *gut t abdominal spot* CRM did not show conserved regions in *D. melanogaster* and may thus not exist.

### 3.5. Transcription Factor Binding Site Analysis of the Abdominal “t spot” CRM Sequence

Previously, we concluded that the genes *dpp*, *wg*, *abd*-*A*, *hh*, and *zen* may collectively induce the *y* gene in spot and shade patterns on the abdomen of *D. guttifera* [1]. To identify putative transcription factors that regulate the *t* gene in this complex abdominal spot pattern, we analyzed the *gut t abdominal spot* CRM sequence using the JASPAR [44] tool. As a result, JASPAR showed three TCF sites and 53 Abd-A sites in the *gut t abdominal spot* CRM, which were distributed relatively evenly across the CRM (Table 1). Putative binding sites for Zen showed 35 sites that appear all over the *gut t abdominal spot* CRM (Table 1). The transcription factor Mothers against dpp (Mad) is a known effector of the *dpp* pathway [45,46]. While JASPAR predicted one Mad site, we did not identify any sites for Cubitus interruptus (Ci), which is a known effector of the *hh* pathway. In addition, JASPAR identified several Engrailed (En) binding sites. En has been shown to be a repressor of pigmentation in the posterior compartment of the wings of *D. biarmipes* [16] and in several drosophilid species of the genus *Samoaia* [47]. The JASPAR tool showed 45 En sites, which may be important for the repression of the stripes into spots (Table 1). However, our RNA in situ hybridization screen did not identify En as a candidate repressor of pigmentation [1]. We also analyzed the “*gut y abdominal spot*” CRM sequences and found that the transcription factor binding site composition between the *y* and *t* abdominal spot CRMs is surprisingly similar [1] (Appendix A), which suggests that the *gut t abdominal spot* CRM may be bound by the same factors that regulate the abdominal *gut y abdominal spot* CRM (Appendix A).

### 3.6. t Is Partially Co-Expressed with y in a Complex Spotted Wing Pattern

The adult wings of *D. guttifera* display 16 pigment spots associated with the wing veins and four intervein shades (Figure 4A). Previously, we showed that *y* expression prefigures the adult wing melanin pattern [2]. Because *t* is co-expressed with *y* on the abdomen of *D. guttifera*, we were curious whether these genes are also identically expressed on the wings of the same species. Our in situ hybridization with the *t* probe on the developing P11 pupal wings only partially correlated with the adult wing melanin pattern. We observed staining in the anterior crossvein, the vein tips of longitudinal veins L2, L3, and L4, and in one of the intervein shades (Figure 4B). Moreover, in our *D. guttifera* transgenic experiments, we identified a 1.9-kb fragment within the third intron of the *t* gene that activated the DsRed reporter in developing pupal wings, precisely foreshadowing the adult wing spot pattern (Figure 3B). We termed this region the *gut t wing spot* CRM. Our data suggest that *t* is co-expressed alongside with *y* in a polka-dotted pattern on the wings of *D. guttifera*.

To identify the candidate inducer of *t* on the wings, we analyzed the activity of the *gut t wing spot* CRM in the genetic background of *D*. *melanogaster.* Previously, we successfully used this strategy to determine that Wg is the inducer of *y* to produce wing spots in adult *D*. *guttifera* [2]. However, the *gut t wing spot* CRM did not activate any reporter expression on the wings of *D. melanogaster*, showing that the *trans*-landscape of *D. melanogaster* does not recognize the *gut t wing spot* CRM, and that factors other than Wg may be inducing *t* in the complex wing spot pattern.

## 4. Discussion

Here we show that the *t* gene is expressed in a multi-spotted pattern in the pupal abdominal epidermis as well as in the pupal wing, each expression pattern being controlled by its own CRM in the *t* gene locus. When we compared these results with our previously published *y* expression data [1], it was apparent that both genes are expressed in nearly identical patterns in the pupal abdomen and wings. Our analysis for putative transcription factors that bind to the abdominal CRMs suggests that the *t* and *y* genes may be regulated by the same *trans*-factors.

The *t* gene is spatially co-expressed with *y* in six rows of spots in the developing abdomen, precisely foreshadowing the complex adult abdominal spot pigment pattern of *D. guttifera*. However, we observe that *t* is expressed at P11, while *y* expression starts at P10 [1]. Consistent with our previous study [24], we detected *t* in *D. guttifera* abdominal spots, but no expression was observed in the dorsal midline of the abdomen. While the *gut t core stripe* CRM shows reporter activation only in abdominal stripes, the *gut y core stripe* CRM activates reporter expression in abdominal stripes as well as in the dorsal midline of the abdomen [1]. Differences in activities of the *t* and *y* CRMs appear to underlie the differences observed in the mRNA expression of these genes on the abdomen, as detected by in situ hybridization. These data suggest that *y* and *t* may have co-evolved to collectively regulate the abdominal spot pattern of *D. guttifera*, whereas the dorsal midline shade pattern may be regulated by *y* alone. Such subtle differences in the expression and regulation of these downstream pigmentation genes may possibly explain the evolution of complex melanin patterns observed in quinaria group species as well as possibly many other flies and even butterflies [48]. 

The sub-fragmentation of the *gut t abdominal spot* CRM reveals a stripe element at its core, suggesting that the *t* expression spots on the abdomen are generated in a similar way to how *y* expression spots are induced on the abdomen [1]. It is likely that the ancestral abdominal pattern was a stripe pattern, which has been repressed later to carve out spots. The ancestral stripe pattern may have been partially repressed over evolutionary time by hypothetical repressors that act longitudinally on the abdomen to silence the spaces between neighboring spots. Although we did not identify any repressors, we showed that the expression patterns of *wg*, *dpp* and *abd-A* closely correlate with the adult spot pattern, suggesting that they may induce *y* on the abdomen in spots [1]. Several species of the quinaria group show signs of such a mechanism at play on their abdomens, such as *Drosophila falleni* [1,35,37] and *Drosophila suboccidentalis* [49]. Considering the similarities in the abdominal expression patterns of the *y* and *t* gene mRNAs, as well as the putative transcription factor binding site composition of the abdominal spot CRMs of the *y* and *t* genes, it is likely that the same *trans*-factors may be collectively regulating *y* and *t* in nearly identical patterns on the abdomen. These data agree with previous studies that suggest that genes with similar mRNA expression patterns and similar functions are likely to participate in the same developmental pathways and are regulated by the same transcription factors [50,51,52,53]. 

We observed that the *t* mRNA signal on the wings shows a rather incomplete correlation with the adult wing spot pattern, while the DsRed reporter pattern driven by the *gut t wing spot* CRM perfectly matches the adult melanin spot pattern. The lower quality of the match of *t* mRNA expression may be due to the fact that *t* is expressed in pupal wings at stage P11, at which the wing cuticle has already hardened, making it more difficult for the probe to reach the epidermal cells of the wing. The fact that the *gut t wing spot* CRM is inactive in transgenic *D. melanogaster* wings suggests that Wg may not be upstream of *t* in activating wing spot expression, while we have shown previously that Wg is upstream of the wing spot CRM of the *y* gene [2]. Thus, other factors must be responsible for the activation of *t* in this polka-dotted wing expression pattern. However, a recent study showed that *t* was upregulated in pigmented areas on the wings of *D. guttifera* caused by the overexpression of *wg* [40]. It is possible that *t* may be indirectly activated by *wg* in wing spots through a transcription factor gene or a signaling molecule that acts downstream of *wg*. A similar mechanism was suggested previously for the spotted activation of the *y* gene in *D. guttifera* by *wg*, where *y* wing spot enhancer-reporter fragments carrying mutations in T-cell factor family protein binding sites did not quench wing spot reporter activity [2]. It is very interesting to note that distinct evolutionary mechanisms may be responsible for shaping the pigment patterns on the wing and abdomen of *D. guttifera*. The wing pigment pattern seems to have emerged through a multistep co-option of multiple gene networks [40]; whereas our data suggest that the abdominal pigment pattern may be assembled by subtle differences in the expression and co-evolution of downstream pigmentation genes [1,24]. Therefore, studies involving *D. guttifera* wing and abdominal melanin patterns may unravel critical insights and progress our understanding of color pattern evolution across non-model species that display a variety of wing and abdominal melanin patterns. 

Future studies utilizing transgenic toolkit gene overexpression, genome editing tools, such as CRISPR-Cas9, and single-fly transcriptomics would help us to understand the role of downstream pigmentation genes and pathways in the formation of melanin patterns. Our study illustrates how similarities and subtle differences in the expression and regulatory architecture enable the co-expression of two pigment-forming genes in complex patterns and extreme diversification of color patterns in the animal kingdom. Because insects display a wide range of color patterns on their wings and bodies [10,28,54,55], *D. guttifera* can serve as a connecting link to understand the evolution of pattern diversity in insects.

## 5. Conclusions

Our transgenic experiments, combined with in situ hybridization data, revealed the CRMs of the *t* gene that drive the spotted expression patterns in the developing wings and abdomen of *D. guttifera*. In this species, the *y* and *t* genes are spatially co-regulated in the abdomen and wings to form complex spotted melanin patterns. Transcription factor binding site analyses suggest that, at least in the abdomen, both the *y* and *t* genes appear to be regulated by a shared set of transcription factors. Our data presented here add to our understanding of how complex color patterns have evolved in animals through the co-regulation of two terminal pigmentation genes.

## Figures and Tables

**Figure 1 genes-14-00304-f001:**
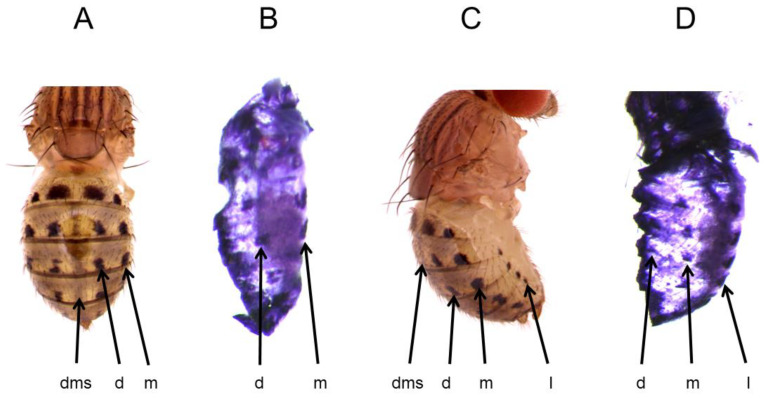
*t* expression foreshadows the three distinct sub-patterns of the adult abdominal spot pattern of *D*. *guttifera*. (**A**). Adult, dorsal view. (**B**). *t* mRNA expression at stage P11, dorsal view. (**C**). Adult, lateral view. (**D**). *t* mRNA expression at stage P11, lateral view. d = dorsal, dms = dorsal midline shade, m = median, and l = lateral.

**Figure 2 genes-14-00304-f002:**
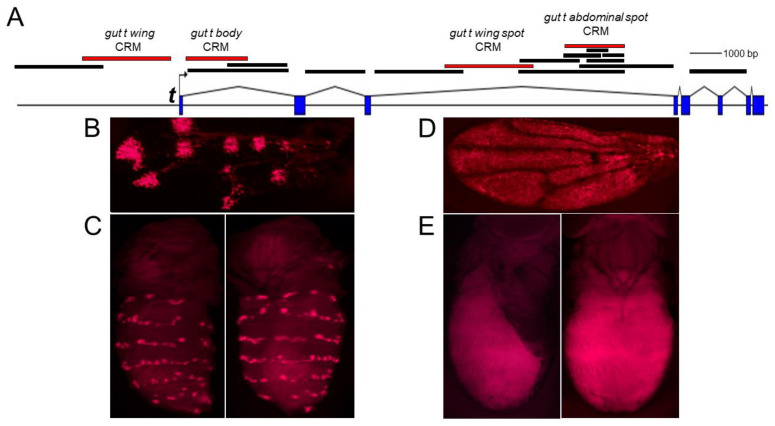
Tissue-specific CRMs of the *t* gene. (**A**). Schematic of the *t* gene locus. The solid bars indicate the DNA fragments of the *t* gene that were analyzed in transgenic *D*. *guttifera* for CRM activity. The red bars indicate the relative position of tissue-specific CRMs. (**B**). Transgenic pupal wing showing the *gut t wing spot* CRM driving DsRed reporter expression in a polka-dotted pattern. (**C**). Transgenic pupal abdomen (left = dorsolateral, right = dorsal) showing the activity of the *gut t abdominal spot* CRM in longitudinal spot rows. (**D**). Transgenic pupal wing showing general *gut t wing* CRM activity. (**E**). Transgenic pupal abdomen (left = lateral, right = dorsal) showing general expression on the body, driven by the *gut t body* CRM.

**Figure 3 genes-14-00304-f003:**
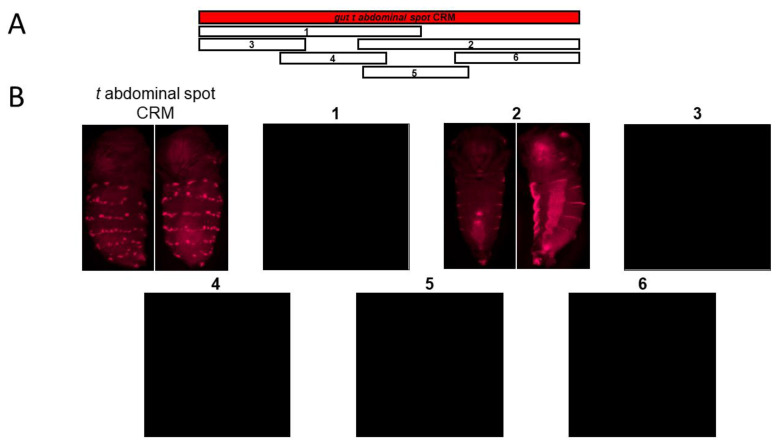
Sub-dividing the *gut t abdominal spot* (**A**) CRM into overlapping sub-fragments revealed horizontal stripes on the abdomen. The *gut t abdominal spot* CRM is shown as a red horizontal rectangular bar. The numbered blocks (1–6) represent the DNA sub-fragments of the *gut t abdominal spot* CRM. The corresponding pupal DsRed expression is shown. Only sub-fragment #2, the *gut t core stripe* CRM (**B**), shows reporter activity in horizontal stripes on the abdomen.

**Figure 4 genes-14-00304-f004:**
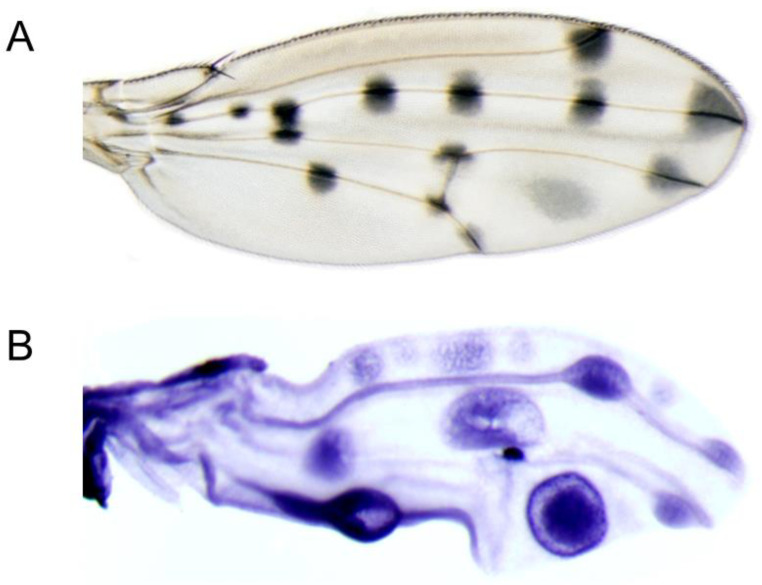
RNA in situ hybridization of the *t* gene on developing pupal wings of *D*. *guttifera*. (**A**). Adult wing showing the melanin pigment pattern. (**B**). P11 pupal wing showing the *t* mRNA expression pattern.

**Table 1 genes-14-00304-t001:** The number of binding sites for candidate toolkit factors revealed by JASPAR analysis of the *gut t abdominal spot* CRM sequence.

EFFECTOR	JASPAR Sites
Wg (TCF/pan)	3
Abd-A	53
Zen	35
Mad (Dpp)	1
Ci (Hh)	0
En	62

## Data Availability

All data are provided within the manuscript.

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
