# Peer review of "The Genetic Mechanisms Underlying the Concerted Expression of the yellow and tan Genes in Complex Patterns on the Abdomen and Wings of Drosophila guttifera"

_genes, 2023, doi:10.3390/genes14020304_

Round 1
Reviewer 1 Report
In this manuscript, Raja et al., examine the coregulation of yellow (y) and tan (t) genes in the abdomen and wing pigmentation of D. guttifera. The study sheds light on the complexity of morphological traits by showing that y and t have differential expression patterns that might be regulated by distinct upstream factors, which remains elusive.
This concise report follow-up to their recent publication, "The regulation of a pigmentation gene in the formation of complex color patterns in Drosophila abdomens" in PLOS ONE. Below are a few minor suggestions for the authors to consider:
1. Keywords are missing.
2. Dorsal and lateral images presented in Figures 1A and C, are used in PLOS One manuscript (Figure 1). The author mentions the reuse of the images.
3. Include the catalog number of the reagents used in the study.
4. For easy readability, consider mentioning 5' and 3' schematics in Figures 2A and 3A.
5. While the pigmentation pattern of male and female D. guttifera are similar, authors have examined y and t gene expression. Authors have any comments?
6. In lines 163-166, the author examined the t gene expression from pupal stages P10 to P15 and observed that t expression begins at P11, whereas y gene expression starts at P10. Images of t expression in different pupal other stages (P10 to P15) would be useful.
Author Response
Reviewer 1 comments
In this manuscript, Raja et al., examine the coregulation of yellow (y) and tan (t) genes in the abdomen and wing pigmentation of D. guttifera. The study sheds light on the complexity of morphological traits by showing that y and t have differential expression patterns that might be regulated by distinct upstream factors, which remains elusive.
This concise report follow-up to their recent publication, "The regulation of a pigmentation gene in the formation of complex color patterns in Drosophila abdomens" in PLOS ONE. Below are a few minor suggestions for the authors to consider:
- Keywords are missing.
We thank the reviewer for the comments. We have added the keywords.
- Dorsal and lateral images presented in Figures 1A and C, are used in PLOS One manuscript (Figure 1). The author mentions the reuse of the images.
We checked permissions from PLOS One and we are allowed to use images that we published, as long as we cite the paper. We have cited our PLOS One paper.
- Include the catalog number of the reagents used in the study.
We thank the reviewer for the comment. This research was performed 5 years back with reagents that were bought more than 5 years ago. We used enzymes, DNA extraction kits, and other reagents from Thermo Fisher and Qiagen. We checked for catalog numbers, and we observed that the catalog numbers changed and therefore did not include them.
- For easy readability, consider mentioning 5' and 3' schematics in Figures 2A and 3A.
We thank the reviewer for the comment. The CRM fragments are part of double-stranded chromosomal DNA, and both strands are in opposite orientations, as is standard. Furthermore, CRMs can activate genes in any orientation. We, therefore, did not include the 5’ and 3’ schematics. However, we would like to point out that there is an arrow next to the “t” at the transcription start site, which indicates the direction of transcription in Figure 2A.
- While the pigmentation pattern of male and female D. guttifera are similar, authors have examined y and t gene expression. Authors have any comments?
- guttifera does not show sexually dimorphic pigment patterns and as expected, we did not observe sex-related differences in y and t expression pattern.
- In lines 163-166, the author examined the t gene expression from pupal stages P10 to P15 and observed that t expression begins at P11, whereas y gene expression starts at P10. Images of t expression in different pupal other stages (P10 to P15) would be useful.
We included a few images showing t expression at other developmental stages. These images are shown in Figure S1.
Reviewer 2 Report
This paper explores the cis-regulatory basis of the pigmentation gene, tan, in Drosophila guttifera. The authors generated a panel of transgenic reporter lines to screen for potential tan enhancers that drives spotted expression. They identified such cis-regulatory modules of tan for expression in spots on the developing pupal abdomen and for spotted wing pattern. Interestingly, both CRMs did not drive reporter expression when introduced into D. melanogaster, which is consistent with the different expression of tan in D mel and D gut and supports a stark change in the transcription factor landscape between the two Drosophila species. The study was well performed and presented.
Only one point to mention, I do not see data supporting the claim that tan gene is co-expressed with the y gene in nearly identical patterns (line 279). Was this known (line270)? Otherwise, please add the data that indicate co-expression.
Author Response
Reviewer 2 comments
This paper explores the cis-regulatory basis of the pigmentation gene, tan, in Drosophila guttifera. The authors generated a panel of transgenic reporter lines to screen for potential tan enhancers that drives spotted expression. They identified such cis-regulatory modules of tan for expression in spots on the developing pupal abdomen and for spotted wing pattern. Interestingly, both CRMs did not drive reporter expression when introduced into D. melanogaster, which is consistent with the different expression of tan in D mel and D gut and supports a stark change in the transcription factor landscape between the two Drosophila species. The study was well performed and presented.
Only one point to mention, I do not see data supporting the claim that tan gene is co-expressed with the y gene in nearly identical patterns (line 279). Was this known (line 270)? Otherwise, please add the data that indicate co-expression.
We thank the reviewer for the comment. It is well known that the downstream pigmentation genes yellow, tan, and dopa decarboxylase are highly conserved and are often co-expressed in pigment patterns of several insect species including Drosophila. Currently, we lack tools to double label both y and t to evaluate their co-expression. However, we found that the expression of both y and t genes is mostly similar and the CRMs drive similar patterns on wings and the abdomen suggesting that they may be co-expressed on the abdomen of D. guttifera.